# Benefits of High-Flow Nasal Cannula Therapy for Acute Pulmonary Edema in Patients with Heart Failure in the Emergency Department: A Prospective Multi-Center Randomized Controlled Trial

**DOI:** 10.3390/jcm9061937

**Published:** 2020-06-21

**Authors:** Dong Ryul Ko, Jinho Beom, Hye Sun Lee, Je Sung You, Hyun Soo Chung, Sung Phil Chung

**Affiliations:** 1Department of Emergency Medicine, Yonsei University College of Medicine, Seoul 06273, Korea; kkdry@yuhs.ac (D.R.K.); WANGTIGER@yuhs.ac (J.B.); emstar@yuhs.ac (S.P.C.); 2Department of Emergency Medicine, Graduate School of Medicine, Kangwon National University, Chuncheon 24289, Korea; 3Department of Research Affairs, Biostatistics Collaboration Unit, College of Medicine, Yonsei University, Seoul 06273, Korea; HSLEE1@yuhs.ac

**Keywords:** high-flow nasal cannula, heart failure, acute pulmonary edema, arterial blood gas, lactate

## Abstract

Heart failure patients with pulmonary edema presenting to the emergency department (ED) require an effective approach to deliver sufficient oxygen and reduce the rate of intubation and mechanical ventilation in the ED; conventional oxygen therapy has proven ineffective in delivering enough oxygen to the tissues. We aimed to identify whether high-flow nasal cannula (HFNC) therapy over time improved the respiratory rate (RR), lactate clearance, and certain arterial blood gas (ABG) parameters, in comparison with conventional oxygen therapy, in patients with cardiogenic pulmonary edema. This prospective, multi-institutional, and interventional study (clinical trial, reference KCT0004578) conducted between 2016 and 2019 included adult patients diagnosed with heart failure within the previous year and pulmonary edema confirmed at admission. Patients were randomly assigned to the conventional or HFNC group and treated with the goal of maintaining oxygen saturation (SpO_2_) ≥ 93. We obtained RR, SpO_2_, lactate levels, and ABG parameters at baseline and 30 and 60 min after randomization. All parameters showed greater improvement with HFNC therapy than with conventional therapy. Significant changes in ABG parameters were achieved within 30 min. HFNC therapy could therefore be considered as initial oxygen therapy. Physicians may consider advanced ventilation if there is no significant improvement in ABG parameters within 30 min of HFNC therapy.

## 1. Introduction

Heart failure (HF) is a serious condition associated with high morbidity and mortality [1,2,3]. Acute pulmonary edema is a major complication of HF, and one of the causes of respiratory failure [3]. Importantly, acute cardiogenic pulmonary edema has an in-hospital mortality rate of approximately 10% and a one-year mortality rate of about 30% [4]. Many patients with HF develop acute pulmonary edema and are admitted to the emergency department (ED) [2,3]. In addition to correction of the underlying causes, the essential treatment for acute respiratory failure due to acute pulmonary edema aims to supply enough oxygen to the tissues and provide optional treatments including diuretics, vasodilators, non-invasive positive pressure ventilation (NIPPV), and endotracheal intubation [5,6,7]. In the ED, conventional oxygen therapy with a nasal cannula or face mask is the most common for patients with dyspnea, owing to high accessibility and convenience for both patients and medical staff [8,9,10]. However, in some patients with acute respiratory failure, this conventional therapy alone proves ineffective in delivering enough oxygen to the tissues [8,9,10]. Several supportive devices have to be used to ensure that the physiological factors and symptoms improve [8,9,10]. To increase the efficacy of treatment, the use of continuous airway positive pressure (CPAP) or noninvasive positive pressure ventilation (NIPPV) may be required prior to endotracheal intubation [8,9,10]. The European Society of Cardiology guidelines for the diagnosis and treatment of acute and chronic heart failure propose the application of noninvasive mechanical ventilation, such as CPAP or NIPPV, to reduce hypercapnia and acidosis and improve the breathing difficulty in dyspneic patients with a respiratory rate of more than 20 breaths/min and acute cardiogenic pulmonary edema (Class IIa recommendation) [1,11]. These abovementioned factors have significantly reduced the need for tracheal intubation and mechanical ventilation [3,12,13]. NIPPV can be used in the form of CPAP or bilevel NIPPV (BiPAP^®^, Respironics, Inc, Murrysville, PA, USA) using face or nasal masks [14]. CPAP maintains constant positive airway pressure throughout the respiratory cycle [14]. In contrast, bilevel NIPPV provides additional inspiratory positive airway pressure and positive end expiratory pressure [14]. These devices are more invasive than conventional oxygen therapy using a nasal cannula or face mask, and can pose limitations for use in ED settings for patients with poor compliance, excessive mucus excretion, altered consciousness, or facial anatomical abnormalities (due to surgery or injury) [5,12,15]. Moreover, CPAP and BiPAP may cause discomfort, leading to failure of treatment and a reduction in cardiac index and venous return in patients with low filling pressure and good ventricular performance [16,17,18].

The use of HFNC therapy may be limited in the patient affected by hypercapnic respiratory failure because HFNC therapy has a minimal effect on reducing the CO_2_ levels by a washout of the anatomical dead space [19]. In recent years, high-flow nasal cannula (HFNC) therapy has been used as an effective approach for delivering sufficient oxygen to patients with acute respiratory failure because this device can potentially generate positive airway pressure, decrease entrainment of ambient air, and reduce the work of breathing [2,19]. Despite patient discomfort with high-flow oxygen applications, the HFNC system can enhance the comfort and tolerability in patients by integrating additional functions for humidification and warming of high-flow oxygen [2,19,20,21]. Based on the aforementioned characteristics, the use of appropriate oxygen therapy can reduce the rate of intubation and mechanical ventilation in the ED [8]

Previous studies have demonstrated that application of HFNC therapy could be a potential therapeutic option in patients with acute respiratory failure [20]. Although several previous studies on HFNC use in patients with acute respiratory failure have been conducted, few studies have demonstrated the clinical effectiveness of HFNC therapy in HF patients with cardiogenic pulmonary edema [2,20,22,23,24]. No international guideline has recommended the use of HFNC therapy in patients with acute cardiogenic pulmonary edema. As research in this field is in the nascent stage, further studies are needed to validate the clinical effectiveness of HFNC therapy in HF patients with cardiogenic pulmonary edema. The prospective study by Makdee et al. demonstrated that HFNC therapy improved the respiratory rate and oxygen saturation in patients (SpO_2_, measured using pulse oximetry), without beneficial effects in ventilation and final outcomes [2]. They also proposed that further studies using objective parameters such as blood gas analysis are required to clarify beneficial effects and generalize the validity of the usefulness of HFNC therapy in HF patients with cardiogenic pulmonary edema [2]. To the best of our knowledge, this is the first prospective, randomized, controlled study involving HF patients with acute pulmonary edema to identify whether HFNC therapy over time improves the respiratory rate (RR), lactate clearance, and parameters in arterial blood gas (ABG, including partial pressure of oxygen (PaO_2_), partial pressure of carbon dioxide (PaCO_2_), pH, and HCO_3_^−^ level), and whether HFNC therapy is superior to conventional oxygen therapy in the early stages of ED admission.

## 2. Experimental Section

### 2.1. Study Design and Patients

This prospective, multi-institutional, and interventional study was performed at two EDs of the Yonsei University College of Medicine affiliated to the Severance Hospital and the Gangnam Severance Hospital between 10 July 2016 and 3 May 2019. The study protocol was reviewed and approved by the institutional review boards of the Severance Hospital (No-1-2016-0030) and Gangnam Severance Hospital (No-3-2016-0063) of the Yonsei University Health System. The trial was registered at cris.nih.go.kr (Clinical Research information Service number (CRiS) Republic of Korea: KCT0004578). Considering the severity of symptoms, written consent was obtained from patients or the legal caregivers at entry into this study. If a patient without written consent recovered from respiratory failure, then a newly written consent was obtained from the patient. The inclusion criteria were: age over 19 years with a diagnosis of HF according to the New York Heart Association (NYHA) classification I–IV and the American Heart Association/European Society of Cardiology (AHA/ESC) guidelines within one year of admission; and acute pulmonary edema confirmed by a chest radiograph at admission. However, the present study excluded patients who were diagnosed with HF at admission. Patients were excluded based on the following criteria: non-cardiogenic pulmonary edema (acute respiratory distress syndrome, ARDS); pneumonia; pregnancy; Glasgow Coma Scale score of 8 points or less; presence of a serious congenital heart condition; on-going dialysis due to renal disease or glomerular filtration rate (GFR) ≤ 30; suspected myocardial infarction (ongoing chest pain, significant change of electrocardiograph, and cardiac enzyme elevation); poor chance of survival due to a pre-existing condition; O_2_ supply alone not being sufficient and the need for immediate invasive trachea management due to severity of symptoms; reluctance to provide consent due to pre-existing conditions or do-not-resuscitate status; cases of transfer from other healthcare institutions following stabilization of symptoms; and inability to provide consent due to the severity of respiratory failure or absence of legal caregivers to authorize the treatment. We performed a multicenter, randomized open-label trial. Patients were randomly assigned to one of the two different treatment groups (conventional oxygen therapy vs. HFNC therapy) using the permuted block of 4 as randomization method.

### 2.2. Intervention

In the conventional oxygen therapy group, oxygen therapy was commenced using a conventional nasal cannula at a flow rate of >2 L/min. The flow rate was continuously adjusted within the conventional nasal cannula or face mask to maintain an SpO_2_ of >93%. In the HFNC group, oxygen therapy was applied using large-bore binasal prongs and a heated humidifier (MR850, Fisher & Paykel Healthcare Limited, Auckland, New Zealand) with a flow rate of 45 L/min and fraction of inspired oxygen (FiO_2_) of 1.0 at initiation (Optiflow, Fisher and Paykel Healthcare, Auckland, New Zealand). The FiO_2_ (from 21% to 100%) and flow rate (up to 60 L/min) in the system were adjusted to maintain an SpO_2_ of >93%. In the study protocol, all patients had to undergo treatment with the assigned modality for at least 60 min. However, according to predetermined criteria of early termination, early intubation and escalation of other devices were allowed if the patients had an intolerable response to the sustained oxygen therapy with either the conventional nasal cannula or HFNC. Early termination criteria included failure to tolerate the therapy (respiratory rate > 35 breaths/min, SpO_2_ < 90%, PaO_2_/FiO_2_ < 200 mmHg, pulse rate > 120 beats/min or a > 30% increase above the baseline and a noninvasively measured pre-intervention mean arterial pressure > 30% higher than that at the baseline or signs of respiratory distress (e.g., tachypnea, use of accessory muscles of respiration, and abdominal paradox), and clinician judgements (when immediate intervention was required due to worsening of the levels of anxiety, agitation, and consciousness compared to those at the pre-intervention timepoint). If one or more of the early termination criteria were met, the oxygen therapy was escalated toward noninvasive ventilation or converted directly to intubation for mechanical ventilation. In addition, all the patients participating in the study were treated with the same standard and concomitant therapy, with a goal of reaching SpO_2_ ≥ 93% and PaO_2_ of 80 mmHg, according to the established treatment guidelines of the AHA for acute pulmonary edema [5]. We obtained RR, SpO_2_, and arterial blood gas analysis (AGBA) data initially and 30 and 60 min after randomization. We also obtained the lactate levels for lactate clearance initially and 60 min after randomization. We analyzed ABG and lactate levels using Stat Profile pHOx Ultra Blood Gas Analyzer (Nova Biomedical, Waltham MA, USA). In this study protocol, we could determine the baseline ejection fraction (EF) from within 6 months of ED admission on the basis of the latest echocardiographic examination. We obtained the baseline EF observed within 1 month preceding the ED from all subjects. In addition, we obtained the EF on emergency echocardiography that was undertaken within 60 min after the intervention in the ED.

### 2.3. Study Outcomes

The primary outcome was to change the objective parameters, including changes in RR, parameters of ABG, and lactate clearance in HF patients with acute pulmonary edema who were assigned into two different treatment groups: HFNC therapy and conventional oxygen therapy. The secondary outcome was to examine the rate of intubation within 24 h after ED admission, the intensive care unit (ICU) admission rate, and all-cause mortality within 28 days of ED admission in each treatment group. We reviewed the medical data from patient discharge and follow-up in the outpatient department that were recorded in the electronic health records (EHRs) during the study period.

### 2.4. Statistical Analysis

The primary focus of the present study was to examine the changes in ABG levels and RR while treating acute pulmonary edema in patients with HF who received either HFNC therapy or conventional oxygen therapy. The number of participants required to identify clinical differences was calculated as 66 participants (33 in each group), based on the minimal detectable differences applied in a previous study that reported PaO_2_ 82.5 ± 17.2 mm Hg for conventional oxygen therapy and PaO_2_ 91.9 ± 7.4 mm Hg for HFNC therapy, with a power of 0.8, and an alpha of 0.05 [24]. We presented demographic and clinical variables and descriptive statistics as medians and interquartile ranges (IQRs), means and standard deviations, percentages, or frequencies, as appropriate. We compared group differences using a chi-squared test or Fisher’s exact test for categorical variables and a *t*-test or Mann–Whitney U test depending on the distribution of continuous variables. We evaluated the serial changes in RR, parameters of ABG, and lactate clearance over time using a linear mixed model as implemented in the MIXED procedure of SAS (version 9.2; SAS Institute, Cary, NC, USA) with restricted maximum likelihood estimation [25]. This analysis uses the observed data from each patient with no imputation for missing data [25]. Fixed effects were treatment (conventional oxygen therapy and HFNC therapy), time of assessment, and the treatment by time interaction [25]. In this model, we analyzed the interaction between the treatment group and time adjusted for baseline RR, lactate, and ABG analysis (ABGA) parameters, as a treatment by time interaction indicates differential changes in RR, lactate, and ABGA parameters over time depending on the treatment [25]. Additionally, we performed post hoc analyses to estimate the time points at which the treatment effects differed between the two groups [25]. In the post hoc analysis, the least square means of two groups were estimated by the MIXED procedure at each time point and compared by the two-sample *t*-test [25]. In this model, we analyzed the interaction between the treatment group and the time-adjusted values for the baseline RR, lactate levels, and ABGA parameters, because the treatment by time interaction indicates differential changes in the RR, lactate levels, and ABGA parameters over time, depending on the treatment [25,26]. Statistical analyses were performed using SAS version 9.2 (SAS Institute Inc., Cary, NC, USA) and MedCalc Statistical Software version 16.4.3 (MedCalc Software bvba, Ostend, Belgium). All statistical tests were two-tailed, and *p* < 0.05 was considered indicative of statistical significance; moreover, 0.05 ≤ *p* < 0.1 was considered to represent a trend toward significance to increase the sensitivity for the detection of potential selection biases.

## 3. Results

### 3.1. Characteristics of Study Subjects

A total of 215 patients were enrolled in this trial. Of these, 69 patients underwent randomization. Two patients withdrew study consent during the trial. Finally, 33 patients were assigned to conventional oxygen therapy, and 34 to HFNC therapy (Figure 1).

Twenty-eight of the subjects were male (41.79%) and the mean age was 76 ± 9 years. Characteristics of patients were similar between the two treatment groups. There were no major between-group differences in the cotreatments, baseline ejection fraction (EF) within one month before ED admission by the latest echocardiographic examination, and EF on emergency echocardiography conducted within 60 min after intervention in the ED. However, systolic blood pressure, blood urea nitrogen (BUN), and brain natriuretic peptide (BNP) levels were statistically higher in the HFNC group (Table 1). The flow rate that resulted in the achievement of the goal of SpO_2_ > 93% was 4.36 ± 3.35 L/min in the conventional O_2_ therapy group, whereas the flow rate and FiO_2_ were 47.58 ± 5.02 L/min and 57.39 ± 14.38, respectively, in the HFNC group.

### 3.2. Study Outcomes

There were significant differences in the RR in the initial, 30 min, and 60 min measurements and in the SpO_2_ at 30 and 60 min between the HFNC and conventional O_2_ therapy groups. With regard to the ABGA parameters, there were significant between-group differences in the PaO_2_ and SpO_2_ at 30 and 60 min (Table 2).

A mixed model analysis of the present study demonstrated that the RR, lactate levels, SPO_2_, and ABG parameters, including PaO_2_, PaCO_2_, and pH, had a significant interaction effect with regard to the treatment group and time. This showed that the HFNC group showed a significant decrease in lactate levels, RR, and PaCO_2_ as well as an increase in the PaO_2_, pH, and SpO_2_ over time. We found changes in RR, lactate levels, SPO_2_, and ABG parameters from baseline to 60 min, depending on the therapy groups (Figure 2 and Figure 3).

We conducted a post hoc analysis to identify the timepoints of the different treatment effects for the two study groups. In the HFNC-treated group, the treatment effects of RR, pH, PaO_2_, PaCO_2_, SPO_2_, and lactate level improved, with statistical significance from the baseline to 60 min over time, indicating greater therapeutic efficacy than in the conventional therapy group. The effects of treatment on the RR, pH, PaCO_2_, and SPO_2_ in the HFNC-treated group indicated significant improvements within 30 min, with greater effectiveness than in the conventional therapy group. However, a statistically significant effect on the HCO_3_ level was not found (Table 3). After adjusting the baseline value, lactate levels revealed a borderline significant trend for the *p*-value, and the statistical significances of differences in other parameters were similar to those identified on post hoc analysis (Table 4).

There was no significant difference in the rates of endotracheal intubation within 24 h between patients receiving conventional oxygen therapy (*n* = 1, 3.0%) and those with HFNC therapy (*n* = 1, 2.94%, *p* = 0.999). All endotracheal intubations were undertaken after the completion of the study interventions. Therefore, patients were not excluded from this study merely due to the need for intubation. In addition, there was no significant difference in ICU admission rate between the conventional oxygen group (*n* = 8, 24.24%) and HFNC group (*n* = 10, 29.41%, *p* = 0.633). There was no significant difference in the overall development of serious adverse events (no cardiac arrest occurred before endotracheal intubation and during the study, and no pneumothorax and pneumomediastinum developed during the study). No significant difference was found in all-cause mortality within 28 days of ED admission between the two groups (no 28-day mortality developed in this study).

## 4. Discussion

We performed the present study to clarify the beneficial effects of HFNC therapy in ED patients with cardiogenic pulmonary edema using objective parameters of RR, lactate levels, SPO_2_, and ABG parameters. To the best of our knowledge, this is the first prospective, randomized, controlled study involving HF patients with acute pulmonary edema to identify whether HFNC therapy improved lactate clearance and objective parameters in arterial blood gas over time. In this study, we observed that several objective parameters including RR, lactate levels, SpO_2_, and ABG parameters (PaCO_2_, pH, PaO_2_, pH, HCO_3_^–^, and SpO_2_) in the HFNC-treated group clinically improved over time compared with the conventional oxygen therapy group in patients with cardiogenic pulmonary edema. Although the present study showed that significant changes in RR, pH, PaO_2_, PaCO_2_, and SPO_2_ in the HFNC-treated group were achieved from the baseline to 60 min, clinically, HFNC therapy improved several parameters including the RR, pH, PaCO_2_, and SPO_2_, from baseline to 30 min after admission and was more effective than conventional oxygen therapy.

The application of HFNC therapy significantly improved oxygenation and decreased the RR in patients with respiratory failure [20]. A retrospective study by Jeong et al. demonstrated that the use of HFNC therapy could significantly increase the PaO_2_, pH, and SpO_2_ and decrease the PaCO_2_ and RR in patients with hypercapnia in the ED [24]. In HFNC therapy, the high flow washes out carbon dioxide from the anatomical dead space of nasopharynx and overcomes resistance against expiratory flow [19,20,27,28]. This produces positive pressure within the nasopharyngeal space that is appropriate for recruiting the collapsed alveoli or for increasing the lung volume (CPAP effect) despite its relatively low pressure compared with closed systems [20,27,28]. HFNC therapy maintains a relatively constant FiO_2_ because of the small difference between the high-flow oxygen that is delivered and the patient’s inspiratory flow [20,27,28]. Patients feel comfortable and the mucociliary function remains good because the HFNC can warm and humidify high flow [19,20,27,28]. Given the physiological benefits of the HFNC, it is clear that HFNC therapy provides a constant FiO_2_ and O_2_ with the nasal cannula, thereby reducing CO_2_ rebreathing, ensuring constant positive pressure, and providing a fresh O_2_ reservoir by washing out the nasopharyngeal dead space. Consequently, the HFNC can maintain sufficient oxygenation by improving the respiratory load and gas exchange in cardiogenic pulmonary edema [23,29]. We found that HFNC therapy could deliver effective oxygenation without major complications or life-threatening adverse events.

Similar to previous studies, the present study also confirmed that HFNC therapy had beneficial effects of change in objective parameters over time in HF patients with cardiogenic edema. In addition, the benefits of the HFNC included greater comfort and tolerability than conventional oxygen therapy because the HFNC system integrates humidification and warming of high-flow oxygen [2,30]. We did not evaluate subjective comfort and clinical tolerability of HFNC in patients. Our study revealed a significant decrease in RR over time after the use of HNFC in patients with cardiogenic pulmonary edema. The application of the HFNC could bridge the gap between conventional oxygen therapy and noninvasive and invasive mechanical ventilation [9].

Makdee et al. suggested that the use of the high-flow nasal cannula in the ED significantly decreased the RR and degree of dyspnea within 30 min [2]. Although the current study demonstrated that significant changes in RR and PaO_2_ in the HFNC-treated group were achieved from the initial timepoint to 60 min, most changes were achieved between baseline and 30 min. Clinically, the HFNC improved several parameters including PaCO_2_, pH, and SpO_2_ between 0 and 30 min and was more effective than conventional oxygen therapy. Considering our results and previous studies, if there is no improvement in the objective parameters such as ABG and RR in 30 min after the use of the HFNC, advanced ventilation devices including noninvasive and invasive mechanical ventilation should be actively considered in HF patients with pulmonary edema.

No serious and life-threatening complications occurred in the HFNC-treated group. However, similar to previous studies, HFNC therapy did not show more benefits compared to conventional oxygen therapy with respect to endotracheal intubation within 24 h, ICU admission rate, and 28-day mortality in HF patients with acute pulmonary edema. Nonetheless, it is noteworthy that the brain natriuretic peptide (BNP) value was statistically higher in the HFNC group than in the conventional O_2_ group in this study. BNP is a hormone that is secreted by the ventricle in response to increased ventricular volume or pressure [31]. The BNP value is elevated when left ventricular systolic function decreases; this elevation is proportional to the severity of HF according to the NYHA classification, and can indicate the long-term prognosis in patients with heart failure [32]. As the pro-BNP value was significantly higher in the HFNC group than in the conventional O_2_ group, the HFNC group may have shown a greater degree of severity of heart failure at the time of ED admission. However, in this study, we cannot exclude the possibility that the pro-BNP level could have affected clinical outcomes such as ICU admission and 28-day mortality. Moreover, it was difficult to conduct a cardiopulmonary exercise test and immediate echocardiography upon ED admission in the emergency setting of our study, and therefore further prospective multicenter studies are required to validate the clinical utility of the HFNC based on the HF severity of patients with cardiogenic pulmonary edema.

There are some limitations to this study. First, although this trial was prospectively conducted in two institutions of Korea, the external validation and generalization of the benefits of HFNC therapy are partially limited. Second, we enrolled adult patients diagnosed with HF according to NYHA I–IV guidelines within one year of admission in this trial. The therapeutic effects of the HFNC are difficult to generalize in the development of acute pulmonary edema in first diagnosed acute and severe cases like acute myocardial infarction. Third, as the present study was conducted by adjusting the FiO2 and flow rate with the goal of maintaining an SpO_2_ > 93%, we recorded FiO_2_ and flow rate that achieved the stated goal. An ABGA was conducted in accordance with the predetermined protocol. Therefore, we could not directly compare the clinical implications between ABGA parameters and FiO_2_ or flow rate in each device. Further studies are needed to validate the impact of ABGA parameters by adjusting the FiO_2_ or flow rate in each device for O_2_. Fourth, as the rate of the HFNC increases, so does the positive pharyngeal pressure. The application of HFNC therapy was maintained below 3 cmH_2_O of positive pharyngeal pressure even at a flow rate of 60 L/min with the mouth open [27,33]. Roca et al. demonstrated that the application of HFNC therapy significantly decreased the median inferior vena cava inspiratory pressure by approximately 20% from the baseline in patients with New York Heart Association Class III heart failure [22]. These changes in the inferior vena cava inspiratory collapse were reversible after HFNC withdrawal [22]. Nevertheless, there were no significant changes in other echocardiographic or clinical variables [22]. Although we obtained the data from echocardiographic studies conducted within 1 month before the ED admission and 60 min after the intervention, to compare the pre- and post-intervention parameters, this study could not completely clarify the positive or negative effects of positive end expiratory pressure (PEEP) of the HFNC in HFNC-treated patients because we were unable to obtain the echocardiographic results immediately before the intervention and after the ED admission. In order to clarify the benefits of the HFNC, further prospective multicenter trials are required to validate the usefulness of HFNC therapy in patients with cardiogenic pulmonary edema.

## 5. Conclusions

Compared with conventional oxygen therapy, HFNC therapy could significantly improve several objective parameters over time such as RR, lactate levels, and ABG reflection of oxygenation and ventilation after ED admission in HF patients with acute pulmonary edema. The application of HFNC therapy could replace conventional O_2_ therapy as initial effective oxygen therapy in patients with cardiogenic pulmonary edema in the ED. In addition, we suggest that physicians consider advanced ventilation devices if there is no significant improvement in several parameters in ABGA after HFNC therapy in patients with cardiogenic pulmonary edema.

## Figures and Tables

**Figure 1 jcm-09-01937-f001:**
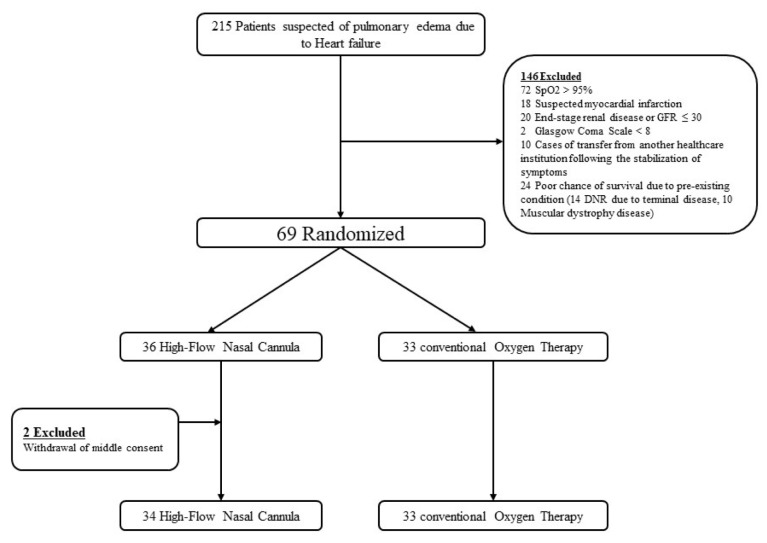
Enrollment and randomization of study participants. GFR, glomerular filtration rate; DNR, do not resuscitate.

**Figure 2 jcm-09-01937-f002:**
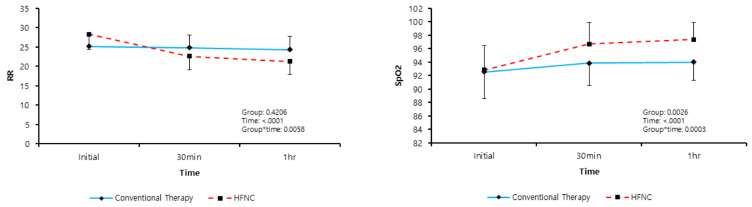
Linear mixed model of changes in respiratory rates (RR) and pulse oximetry (SpO_2_). The graphs show interaction effects between time and treated group using the linear mixed model (Group * Time). HFNC: high-flow nasal cannula.

**Figure 3 jcm-09-01937-f003:**
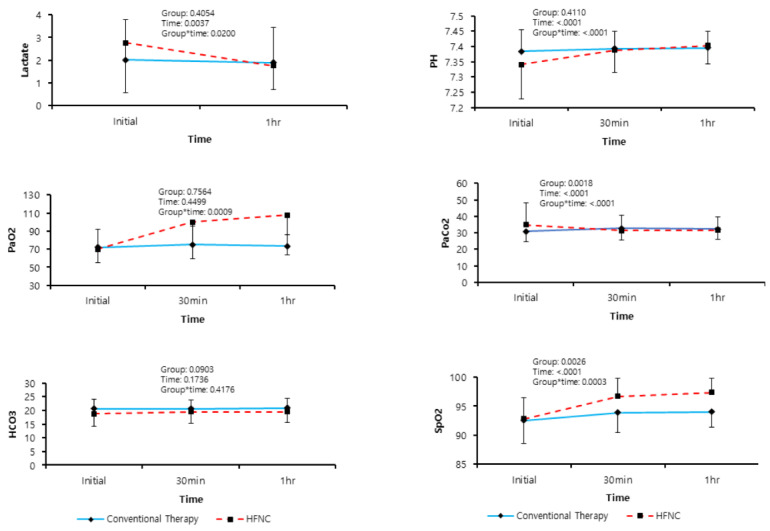
Linear mixed model of changes in arterial blood gas analysis parameters. The graphs show interaction effects between time and treated group using linear mixed model (Group * Time), PaO_2_: partial pressure of oxygen, PaCO_2_: partial pressure of carbon dioxide, SpO_2_: peripheral oxygen saturation, HFNC: high-flow nasal cannula.

**Table 1 jcm-09-01937-t001:** Baseline characteristics of patients.

Variable	Total (*n* = 72)	Conventional O_2_ Therapy Group	High-Flow Nasal Cannula Group	*p*-Value
	Mean ± SD or *n* (%)	(*n* = 33, 49.3%)	(*n* = 34, 50.7%)	
Age (years)	76 ± 9	76 ± 9	77 ± 8	0.530
Male sex, number (%)	28(41.79)	13(39.39)	15(44.12)	0.695
Comorbidity (%)				
Hypertension	40(59.70)	18(54.55)	22(64.71)	0.396
Diabetes mellitus	25(37.31)	11(33.33)	14(41.18)	0.506
Cardiovascular disease	9(13.43)	6(18.18)	3(8.82)	0.305
Chronic kidney disease	4(5.97)	2(6.06)	2(5.88)	0.999
Systolic blood pressure (mmHg)	148.96 ± 33.81	140.24 ± 22.41	157.47 ± 40.61	0.035 *
Diastolic blood pressure (mmHg)	86.67 ± 24.98	81.33 ± 13.71	91.85 ± 31.78	0.084 *
Heart rate (bpm)	95.51 ± 22.61	92.61 ± 19.33	98.32 ± 25.37	0.304
Body temperature (℃)	36.10 ± 3.26	35.43 ± 4.54	36.75 ± 0.56	0.106
Laboratory data **				
White blood cell count (μL)	10855.22 ± 6803.40	10052.42 ± 5724.55	11634.41 ± 7715.25	0.345
Hemoglobin (g/dL)	11.66 ± 2.17	11.34 ± 1.99	11.976 ± 2.32	0.235
Hematocrit (%)	36.32 ± 6.73	34.99 ± 6.26	37.62 ± 6.99	0.109
Platelet count (10^3^/μL)	216.97 ± 86.38	210.18 ± 92.31	223.56 ± 81.04	0.530
Blood urea nitrogen (mg/dL)	30.90 ± 17.85	25.86 ± 12.86	35.80 ± 20.66	0.021 *
Creatinine (mg/dL)	1.59 ± 1.07	1.40 ± 0.90	1.79 ± 1.20	0.139
Albumin (g/dL)	3.65 ± 0.53	3.66 ± 0.53	3.64 ± 0.53	0.881
Aspartate aminotransferase (IU/L)	38.27 ± 21.99	38.76 ± 20.15	37.79 ± 23.94	0.859
Alkaline phosphatase (IU/L)	26.73 ± 25.84	27.73 ± 27.07	25.77 ± 24.96	0.759
Total bilirubin (mg/dL)	1.14 ± 1.40	1.13 ± 1.31	1.15 ± 1.51	0.962
Sodium (mmol/L)	137.40 ± 5.44	137.55 ± 6.04	137.27 ± 4.87	0.835
Potassium (mmol/L)	4.44 ± 0.80	4.15 ± 0.75	4.73 ± 0.76	0.003 *
Chloride (mmol/L)	102.46 ± 5.97	103.15 ± 6.79	101.79 ± 5.07	0.351
Creatine kinase (U/L)	114.33 ± 87.24	105.73 ± 64.76	122.68 ± 104.92	0.428
Creatine kinase myocardial bandisoenzyme (mcg/L)	4.19 ± 3.17	3.35 ± 2.72	5.02 ± 3.38	0.030 *
Troponin-I (mcg/L)	0.0842 ± 0.099	0.0674 ± 0.081	0.101 ± 0.112	0.169
Pro-brain natriuretic peptide (pg/mL)	8383.51 ± 14132.58	2662.59 ± 2092.40	13936.17 ± 18185.67	0.001 *
PaO_2_/FiO_2_ (initial)	337.4 3± 82.50	342.43 ± 94.17	332.58 ± 70.45	0.371
Echocardiography—previous ED visit				
Ejection fraction (%)	46.10 ± 15.18	44.52 ± 15.30	47.65 ± 15.13	0.403
Valve disease	17(25.37)	7(21.21)	10(29.41)	0.441
Furosemide	67(100.00)	33(100.00)	34(100.00)	0.999
Dobutamine	48(71.64)	25(75.76)	23(67.65)	0.462

* *p* < 0.05, ED: emergency department, FiO_2_: fraction of inspired oxygen (FiO_2_), PaO_2_: partial pressure of oxygen. ** reference range, Appendix A.

**Table 2 jcm-09-01937-t002:** Outcomes.

Variable	Total (*n* = 72)	Conventional O_2_ Therapy Group	High-Flow Nasal Cannula Group	*p*-Value
	Mean ± SD or *n* (%)	(*n* = 33, 49.3%)	(*n* = 34, 50.7%)	
**Respiratory rate (bpm)**				
Initial	26.78 ± 3.99	25.18 ± 3.51	28.32 ± 3.86	0.001 *
30 min	23.75 ± 3.50	24.85 ± 3.19	22.68 ± 3.49	0.010 *
60 min	22.79 ± 3.72	24.30 ± 3.55	21.32 ± 3.32	0.001 *
**SpO_2_ (%)**				
Initial	91.41 ± 5.89	92.55 ± 3.78	90.31 ± 7.29	0.120
30 min	95.69 ± 3.31	94.15 ± 3.26	97.18 ± 2.65	<0.001 *
60 min	95.94 ± 3.27	94.12 ± 3.25	97.71 ± 2.14	<0.001 *
**Arterial Blood Gas Analysis**				
pH, initial	7.36 ± 0.09	7.38 ± 0.07	7.34 ± 0.11	0.063
pH, 30 min	7.39 ± 0.07	7.39 ± 0.06	7.39 ± 0.07	0.788
pH, 60 min	7.40 ± 0.06	7.40 ± 0.06	7.40 ± 0.06	0.595
PaO_2_, initial	70.86 ± 17.32	71.91 ± 19.78	69.84 ± 14.79	0.629
PaO_2_ 30 min	87.79 ± 34.46	75.23 ± 19.87	99.98 ± 41.00	0.003 *
PaO_2_ 60 min	90.62 ± 36.79	73.25 ± 13.02	107.47 ± 44.15	<0.001 *
PaCO_2_, initial	32.85 ± 10.44	30.89 ± 6.18	34.76 ± 13.17	0.129
PaCO_2_, 30 min	31.97 ± 8.21	32.61 ± 7.13	31.35 ± 9.20	0.532
PaCO_2_, 60 min	31.91 ± 7.22	32.30 ± 6.22	31.54 ± 8.14	0.670
SpO_2_ (%), initial	92.69 ± 3.79	92.55 ± 4.01	92.83 ± 3.63	0.765
SpO_2_, 30 min	95.30 ± 3.55	93.86 ± 3.38	96.71 ± 3.17	0.001 *
SpO_2_, 60 min	95.71 ± 3.07	93.99 ± 2.64	97.38 ± 2.51	<0.001 *
**Lactate (mmol/L)**				
Initial	2.39 ± 2.02	2.01 ± 1.78	2.77 ± 2.20	0.126
60 min	1.82 ± 1.31	1.89 ± 1.55	1.75 ± 1.04	0.666
**Echocardiography After ED visit**				
Ejection fraction (%)	40.15 ± 13.12	40.36 ± 15.23	39.94 ± 10.92	0.896
Valve disease	18(26.87)	8(24.24)	10(29.41)	0.633
Intubation	2(2.99)	1(3.03)	1(2.94)	0.999
ICU admission	18(26.87)	8(24.24)	10(29.41)	0.633

* *p* < 0.05, PaO_2_: partial pressure of oxygen, PaCO_2_: partial pressure of carbon dioxide, SpO_2_: peripheral oxygen saturation.

**Table 3 jcm-09-01937-t003:** Post hoc analysis to estimate the time points at which the treatment effects differed between the 2 groups.

Group Post-Hoc *p*-Value		Time Post-Hoc *p*-Value			Group * Time Post Hoc	
Conventional vs. HFNC		Conventional vs. HFNC			*p*-Value	
**Respiratory Rate**						
Initial	0.001 *	Initial vs. 30 min	0.250	<0.001 *	C vs. HFNC and Initial vs. 30 min	<0.001 *
30 min	0.010 *	Initial vs. 1 h	0.037 *	<0.001 *	C vs. HFNC and Initial vs. 1 h	<0.001 *
1 h	0.001 *	30 min vs. 1 h	0.064	<0.001 *	C vs. HFNC and 30 min vs. 1 h	0.051
**Arterial Blood Gas Analysis (ABGA), pH**						
Initial	0.064	Initial vs. 30 min	0.327	<0.001 *	C vs. HFNC and Initial vs. 30 min	0.002 *
30 min	0.788	Initial vs. 1 h	0.330	<0.001 *	C vs. HFNC and Initial vs. 1 h	0.003 *
1 h	0.595	30 min vs. 1 h	0.714	0.043 *	C vs. HFNC and 30 min vs. 1 h	0.241
**ABGA, PaCO_2_**						
Initial	0.130	Initial vs. 30 min	0.073	0.001 *	C vs. HFNC and Initial vs. 30 min	<0.001 *
30 min	0.532	Initial vs. 1 h	0.250	0.009	C vs. HFNC and Initial vs. 1 h	0.008
1 h	0.670	30 min vs. 1 h	0.648	0.779	C vs. HFNC and 30 min vs. 1 h	0.602
**ABGA, PaO_2_**						
Initial	0.629	Initial vs. 30 min	0.520	<0.001 *	C vs. HFNC and Initial vs. 30 min	<0.001 *
30 min	0.003 *	Initial vs. 1 h	0.809	<0.001 *	C vs. HFNC and Initial vs. 1 h	<0.001 *
1 h	<0.001 *	30 min vs. 1 h	0.518	0.015 *	C vs. HFNC and 30 min vs. 1 h	0.030 *
**ABGA, HCO_3_**						
Initial	0.056	Initial vs. 30 min	0.871	0.089	C vs. HFNC and Initial vs. 30 min	0.188
30 min	0.220	Initial vs. 1 h	0.469	0.055	C vs. HFNC and Initial vs. 1 h	0.398
1 h	0.106	30 min vs. 1 h	0.289	0.831	C vs. HFNC and 30 min vs. 1 h	0.543
**ABGA, SpO_2_**						
Initial	0.765	Initial vs. 30 min	0.020 *	<0.001 *	C vs. HFNC and Initial vs. 30 min	0.001 *
30 min	0.001 *	Initial vs. 1 h	0.008 *	<0.001 *	C vs. HFNC and Initial vs. 1 h	<0.001 *
1 h	<0.001 *	30 min vs. 1 h	0.715	0.062	C vs. HFNC and 30 min vs. 1 h	0.288
**Lactate**						
Initial	0.126	Initial vs. 60 min	0.661	<0.001 *	C vs. HFNC and Initial vs. 60 min	0.020 *
60 min	0.664					

* *p* < 0.05, C: conventional therapy group, HFNC: high-flow nasal cannula group, ABGA: arterial blood gas analysis, PaO_2_: partial pressure of oxygen, PaCO_2_: partial pressure of carbon dioxide, SpO_2_: peripheral oxygen saturation.

**Table 4 jcm-09-01937-t004:** Analysis of the interaction between the treatment group and the time-adjusted values for the baseline.

	Conventional O_2_ Therapy Group	HFNC Group	*p*-Value ^†^
	Estimated Mean (SE)	Estimated Mean (SE)	
**Respiratory Rate**			
30 min–Initial	−0.644(0.274)	−5.346(0.27)	< 0.001 *
1 h–Initial	−1.189(0.386)	−6.999(0.38)	<0.001 *
1 h–30 min	−0.546(0.29)	−1.353(0.285)	0.051
**Arterial Blood Gas Analysis (ABGA), pH**			
30 min–Initial	0.019(0.006)	0.038(0.006)	0.027 *
1 h–Initial	0.021(0.007)	0.053(0.007)	0.004 *
1 h–30 min	0.003(0.007)	0.015(0.01)	0.241
**ABGA, PaCO_2_**			
30 min–Initial	1.003(0.766)	−2.712(0.755)	0.001 *
1 h–Initial	0.688(0.885)	−2.521(0.872)	0.013 *
1 h–30 min	−0.315(0.687)	0.191(0.677)	0.602
**ABGA, PaO_2_**			
30 min–Initial	3.6(5.143)	29.866(5.067)	0.001 *
1 h–Initial	1.622(5.444)	37.355(5.363)	<0.001 *
1 h–30 min	−1.979(3.042)	7.488(2.997)	0.030 *
**ABGA, HCO_3_**			
30 min–Initial	0.195(0.394)	0.47(0.388)	0.625
1 h–Initial	0.568(0.372)	0.543(0.366)	0.963
1 h–30 min	0.373(0.349)	0.074(0.344)	0.543
**ABGA, SpO_2_**			
30 min–Initial	1.223(0.453)	3.957(0.446)	<0.001 *
1 h–Initial	1.354(0.353)	4.621(0.347)	<0.001 *
1 h–30 min	0.13(0.355)	0.665(0.35)	0.288
**Lactate**			
60 min–Initial	−0.342(0.18)	−0.801(0.177)	0.076

* *p* < 0.05, HFNC: high-flow nasal cannula group, ABGA: arterial blood gas analysis, RR: respiratory rate, PaO_2_: partial pressure of oxygen, PaCO_2_: partial pressure of carbon dioxide, SpO_2_: peripheral oxygen saturation, ^†^ Adjustment; baseline value of parameter. SE, standard error.

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
