# Peer review of "Benefits of High-Flow Nasal Cannula Therapy for Acute Pulmonary Edema in Patients with Heart Failure in the Emergency Department: A Prospective Multi-Center Randomized Controlled Trial"

_jcm, 2020, doi:10.3390/jcm9061937_

Round 1

Reviewer 1 Report

This open-label randomized controlled trial aims to assess respiratory parameters and blood gas variables in patients with ACPE treated with HFNT vs conventional oxygen therapy. HFNT appears to improve RR, oxygenation and blood gas variables in the short term. Overall, I would suggest the Author to add to their discussion, commenting more in depth their results. 

Specific comments:

  1. Introduction
    1. Page 2 line 47 - Oxygen is addressed as the most common treatment used in ED. By reading the introduction it appears as referred to the most common treatment used for patients with ACPE, or was it meant to be overall? Furthermore, could the Author review the references provided as these are on HFNT rather than on the use of oxygen for ACPE?
    2. Page 2 line 52-62 - The authors describe the use of CPAP/NIV in the context of ACPE as a technique to provide ventilatory support prior to intubation. Also the Authors comment on the discomfort c However, CPAP and NIV may also reduce the need for intubation in this setting. Can the Author comment on this?
    3. Minor comments: 
      1. page 2 line 65 entrapment should be entrainment
      2. page 2 line 66 breathing work should be work of breathing
      3. page 2 line 69 - findings should be characteristics 
      4. page 2 line 72 - Authors mention previous studies then reference is only to Roca et al. 
  2. Methods
    1. Page 3 Intervention line 110 - the first line in the paragraph describing the study is actually the study design, as such I would suggest to move it in the section before. 
    2. Patients were randomly assigned to intervention or control group and had to continue with the treatment they were assigned to for at least 60 minutes, unless intubation was required. Was cross-over allowed in case of deterioration, or were patients allowed to be transitioned to treatment with CPAP or NIV? If yes, this should be described in the methods section then commented in results. If not could the Authors clarify their choice for not allowing escalation to treatment to CPAP in the context of ACPE?
    3. Study outcome - the outcome is not to compare, but are the changes per se. 
  3. Results
    1. Page 5 line 166-167 Can the Author review the first sentence as it appears inconclusive? 
    2. Line 169 - 171: I believe this is not really needed in here. The exclusion criteria are stated in the Methods section and the breakdown is in figure 2
    3. Figure 1 - I believe this figure is not really needed, and otherwise would apply to the methods section rather than the results
    4. Table 1 - I would suggest to add the normal range for biochemistry results in the Authors' Institution lab for further clarification. 
    5. Can the Author add a table comparing baseline RR, SpO2 and ABG results in HFNT and oxygen group. Reading table 2, it appears that subjects in the HFNT group were more severe than those in the oxygen group. RR was significantly higher, SpO2 tended to be lower, pH tended to be more acidotic, etc. 
    6. Page 6 - study outcomes section
      1. Can the Authors clarify in the main manuscript the effects of HFNT (i.e. provide the difference in RR, SpO2 and ABG variables)? It is difficult to follow the flow having to read the tables to actually appreciate the results, which appear more discussed rather than presented in this part of the results section. 
      2. Line 186 - 189 this is a conclusion rather than a presentation of the result, so I would suggest to move this in the discussion. 
    7. Can the Authors provide details re treatment? Specifically, I think it would be important for the Author to clarify the average flow-rate/Fio2 with HFNT and the flow-rate and device for O2 during the study period. Currently, only the initial setting are provided. This is of the utmost importance to understand the impact of the results on blood gas analysis as well.  
  4. Discussion
    1. Line 269-274 - The Author comment on the need for further studies to assess the effect on longer term outcome, ICU admission and ETI rate. However, this is as a consequence of the lack of difference in this outcomes in their study, which, however, was not powered or designed to assess such outcomes. 
    2. Line 283 - Can the Author clarify? In the result section it was stated the the echo was within 6 months. Also, an echo 60 minutes after intervention is discussed, but in the result section only the baseline echo on admission is presented. Can the Author clarify and comment? If the echo in the ED was performed after the intervention, this should not be presented as a baseline characteristic in Table 1 as it would be confusing. Furthermore, can the Author comment on how the HFNT could influence echocardiography results. Finally, the echo (both retrieval of the most recent one, and performing in ED) is not discussed in the methods section.  
    3. The Authors mention the role of HFNT as a bridge prior to other noninvasive forms of support or to invasive ventilation. Can the Author discuss this more in depth? 
    4. I would like to read a more in depth discussion on how the proposed physiological mechanisms of HFNT lead to a beneficial effect and support the Author's results. 

Author Response

** Please, check attached file **

Response to reviewers’ comments

Benefits of high flow nasal cannula therapy for acute pulmonary edema in patients with heart failure in the emergency department: a prospective multi-center randomized controlled trial

Comments and Suggestions for Authors

This open-label randomized controlled trial aims to assess respiratory parameters and blood gas variables in patients with ACPE treated with HFNT vs conventional oxygen therapy. HFNT appears to improve RR, oxygenation and blood gas variables in the short term. Overall, I would suggest the Author to add to their discussion, commenting more in depth their results. 

Response: Thank you. We appreciate your kind comments and valuable suggestions with regard to our manuscript.

Specific comments:

  1. Introduction
    1. Page 2 line 47 - Oxygen is addressed as the most common treatment used in ED. By reading the introduction it appears as referred to the most common treatment used for patients with ACPE, or was it meant to be overall? Furthermore, could the Author review the references provided as these are on HFNT rather than on the use of oxygen for ACPE?

Response: Thank you for your kind comment. The initial paragraph in this section presented the general treatment that is commonly used in patients with dyspnea (lines 46-49). In the introduction, we presented the general therapeutic modalities that are used in acute cardiogenic pulmonary edema and also described the advantages and disadvantages of these modalities (lines 50-67). In the next paragaph, we presented the therapeutic benefits and the need for HFNC therapy in patients with acute cardiogenic pulmonary edema (lines 68-75). As suggested, in consideration of the logical flow of the text, we have presented the use of HFNC therapy in acute cardiogenic pulmonary edema and have placed it before the statement on the objective of this study.

“Although several previous studies on HFNC use in patients with acute respiratory failure have been conducted, few studies have demonstrated the clinical effectiveness of HFNC in HF patients with cardiogenic pulmonary edema [2,20,22-24]. No international guideline has recommended the use of HFNC in patients with acute cardiogenic pulmonary edema. As research in this field is in the nascent stage, further studies are needed to validate the clinical effectiveness of HFNC in HF patients with cardiogenic pulmonary edema.”

“Although there are only a few prospective studies on HFNC for the treatment of acute pulmonary edema in patients with HF, The prospective study by Makdee et al. demonstrated that HFNC improved the respiratory rate and oxygen saturation in patients (SpO2, measured using pulse oximetry), albeit without beneficial effects on ventilation and final outcomes [2].”

  1. Page 2 line 52-62 - The authors describe the use of CPAP/NIV in the context of ACPE as a technique to provide ventilatory support prior to intubation. Also the Authors comment on the discomfort c However, CPAP and NIV may also reduce the need for intubation in this setting. Can the Author comment on this?

Response: Thank you for your kind feedback. We have added this point from the manuscript accordingly.

“The european society of cardiology guidelines for the diagnosis and treatment of acute and chronic heart failure proposed the application of noninvasive mechanical ventilation, such as CPAP or NIPPV, to reduce hypercapnia and acidosis and improve the breathing difficulty in dyspneic patients with a respiratory rate of more than 20 breaths/min and acute cardiogenic pulmonary edema (Class IIa recommendation) [1,11]. These abovementioned factors have significantly reduced the need for tracheal intubation and mechanical ventilation [3,12,13].”

In lines 57-62, there is content that explains the disadvantages of CPAP and NIPPV as follows:

“These devices are more invasive than conventional oxygen therapy using a nasal cannula or face mask, and can pose limitations for use in ED settings for patients with poor compliance, excessive mucus excretion, altered consciousness, or facial anatomical abnormalities (due to surgery or injury) [5,12,15]. Moreover, CPAP and BiPAP may cause discomfort leading to failure of treatment, decrease in cardiac index, and venous return in patients with low filling pressure and good ventricular performance [16-18].”

  1. Minor comments: 
    1. page 2 line 65 entrapment should be entrainment

Response: Thank you. We have accordingly revised the manuscript (entrapment > entrainment).

  1. page 2 line 66 breathing work should be work of breathing

Response: Thank you. We have accordingly revised the manuscript (breathing work > work of breathing).

  1. page 2 line 69 - findings should be characteristics 

Response: Thank you. We have accordingly revised the manuscript (findings > characteristics).

  1. page 2 line 72 - Authors mention previous studies then reference is only to Roca et al. 

Response: Thank you. We have accordingly revised the manuscript

“Although several previous studies on HFNC use in patients with acute respiratory failure have been conducted, few studies have demonstrated the clinical effectiveness of HFNC in HF patients with cardiogenic pulmonary edema [2,20,22-24].”

  1. Methods

  1. Page 3 Intervention line 110 - the first line in the paragraph describing the study is actually the study design, as such I would suggest to move it in the section before. 

Response: Thank you. We have accordingly revised the manuscript.

Patients were randomly assigned to intervention or control group and had to continue with the treatment they were assigned to for at least 60 minutes, unless intubation was required. Was cross-over allowed in case of deterioration, or were patients allowed to be transitioned to treatment with CPAP or NIV?

If yes, this should be described in the methods section then commented in results. If not could the Authors clarify their choice for not allowing escalation to treatment to CPAP in the context of ACPE?

Response: Thank you for this feedback. In our study, we determined the criteria for early termination based on modifications to the guidelines; early intubation and escalation of other devices was allowed if the patients had an intolerable response to continuous oxygen therapy with either the conventional nasal cannula or HFNC. However, only 2 endotracheal intubations were undertaken after the study interventions had been completed. There was no escalation reported for the other devices (CPAP or NIPPV).

To clarify this point, we have accordingly included the predetermined criteria for early termination in the revised manuscript.

‘However, according to predetermined criteria of early termination, early intubation and escalation of other devices was allowed if the patients had an intolerable response to the sustained oxygen therapy with either the conventional nasal cannula or HFNC. The early termination criteria included the failure to tolerate the therapy (respiratory rate >35 breaths/min, SpO2 <90%, PaO2/FiO2 <200 mmHg, pulse rate >120 beats/min or a >30% increase above the baseline, and a noninvasively measured pre-intervention mean arterial pressure that was >30% higher than that at the baseline, or signs of respiratory distress (e.g. tachypnea, use of accessory muscles of respiration, and abdominal paradox), and clinician judgements (when immediate intervention is required due to worsening of the levels of anxiety, agitation, and consciousness compared to those at the pre-intervention timepoint). If one or more of the early termination criteria were met, the oxygen therapy was escalated toward noninvasive ventilation or converted directly to intubation for mechanical ventilation.”

Results

“There was no significant difference in the rates of endotracheal intubation within 24 hours in patients receiving conventional oxygen therapy (n=1, 3.0%) and those undergoing HFNC (n=1, 2.94%, p=0.999). All endotracheal intubations were undertaken after the completion of the study interventions. Therefore, patients were not excluded from this study merely due to the need for intubation.”

  1. Study outcome - the outcome is not to compare, but are the changes per se. 

Response: Thank you. We have accordingly revised the manuscript.

  1. Results
    1. Page 5 line 166-167 Can the Author review the first sentence as it appears inconclusive? 

Response: Thank you. Accordingly, we have deleted this sentence from the manuscript.

  1. Line 169 - 171: I believe this is not really needed in here. The exclusion criteria are stated in the Methods section and the breakdown is in figure 2

Response: Thank you. As suggested, we have deleted this content from the manuscript to avoid redundancy.

  1. Figure 1 - I believe this figure is not really needed, and otherwise would apply to the methods section rather than the results

Response: Thank you. We have accordingly deleted the figure from the manuscript.

Table 1 - I would suggest to add the normal range for biochemistry results in the Authors' Institution lab for further clarification. 

Response: Thank you for this suggestion. We added the normal range of the biochemical parameters to the tables in manuscript, but this made the table too complex and too large to facilitate readability. Therefore, we have accordingly included the normal range values for the results of biochemical analysis in the supplementary materials for this manuscript.

  1. Can the Author add a table comparing baseline RR, SpO2 and ABG results in HFNT and oxygen group. Reading table 2, it appears that subjects in the HFNT group were more severe than those in the oxygen group. RR was significantly higher, SpO2 tended to be lower, pH tended to be more acidotic, etc. 

Response: Thank you for this helpful comment. We completely agree with your comment. This is the first prospective, randomized, controlled study involving HF patients with acute pulmonary edema, to identify whether HFNC over time improves the respiratory rate (RR), lactate clearance, and parameters on the arterial blood gas (ABG) analysis, including partial pressure of oxygen (PaO2), PaCO2, pH, and HCO3- level. Furthermore, we ascertained whether HFNC is superior to conventional oxygen therapy in the early stages of ED admission. In this study, the baseline values of RR, SPO2, and parameters in arterial blood gas (ABG) analysis were recorded as the initial values on ED admission. As we could not determine the specific timepoint of similar disease status in all subjects before ED admission, we did not consider the changes of parameters from before to after ED admission in the study design. We did not analyze the pre-admission details of subjects from before the ED admission as the values of several parameters before the ED admission reflected the absence of pathological acute cardiogenic pulmonary edema in these patients. The differences in the baseline data may be caused by differences in the disease status in symptomatic patients with acute cardiogenic pulmonary edema after ED admission. We understand that you are concerned that the differences in baseline data between the two groups may have contributed to errors in the main analysis. We have discussed this point with our statisticians, and have subsequently added the results of the linear mixed model analysis after adjusting the baseline values to those shown in Table 4. In the analysis that was conducted after adjusting the baseline value, the lactate revealed a borderline significant trend for the p-value, whereas the statistical significances of the other parameters were similar to those obtained in the post-hoc analysis.

Statistical analysis

“In this model, we analyzed the interaction between the treatment group and the time adjusted values for the baseline RR, lactate levels, and ABGA parameters, because the treatment by time interaction indicates differential changes in the RR, lactate levels, and ABGA parameters over time, depending on the treatment [25,26].

All statistical tests were two-tailed, and p<0.05 was considered indicative of statistical significance; moreover, 0.05≤p<0.1 was considered to represent a trend toward significance to increase the sensitivity for the detection of potential selection biases.”

Results

“After adjusting the baseline value, lactate levels revealed a borderline significant trend for the p-value, and the statistical significance of differences in other parameters were similar to those identified on post hoc analysis (Table 4).”

  1. Page 6 - study outcomes section
    1. Can the Authors clarify in the main manuscript the effects of HFNT (i.e. provide the difference in RR, SpO2 and ABG variables)? It is difficult to follow the flow having to read the tables to actually appreciate the results, which appear more discussed rather than presented in this part of the results section. 

Response: Thank you for this feedback. We have accordingly revised the manuscript.

3.2. Study outcomes

“There were significant differences in the RR from the initial, 30 min, and 60 min measurements and in the SpO2 at 30 and 60 min between the HFNC and conventional O2 therapy groups. With regard to the ABGA parameters, there were significant between-group differences in the PaO2 and SpO2 at 30 and 60 min (Table 2).

A mixed model analysis of the present study demonstrated that the RR, lactate levels, SPO2, and ABG parameters, including PaO2, PaCO2, and pH, had a significant interaction effect with regard to the treatment group and time. This showed that the HFNC group had a significant decrease in lactate levels, RR, and PaCO2 as well as an increase in the PaO2, pH, and SpO2 over time. We found changes in RR, lactate levels, SPO2, and ABG parameters from baseline to 60 minutes, depending on the therapy groups (Figures 3 and 4).

We conducted a post hoc analysis to identify the timepoints of the different treatment effects for the two study groups. In the HFNC-treated group, the treatment effects of RR, pH, PaO2, PaCO2, SPO2, and lactate level improved with statistical significance from the baseline to 60 minutes over time and indicated greater therapeutic efficacy than in the conventional therapy group. The effects of treatment on the RR, pH, PaCO2, and SPO2 in the HFNC-treated group indicated significant improvement within 30 min and were more effective than in the conventional therapy group. However, a statistically significant effect on the HCO3 level was not found (Table 3). After adjusting the baseline value, lactate levels revealed a borderline significant trend for the p-value, and the statistical significance of differences in other parameters were similar to those identified on post hoc analysis (Table 4).”

Discussion

“Although the present study showed that significant changes in RR, pH, PaO2, PaCO2, and SPO2 in the HFNC-treated group were achieved from the baseline to 60 minutes, clinically, HFNC improved several parameters, including the RR, pH, PaCO2, and SPO2, from the baseline to 30 minutes after admission and was more effective than conventional oxygen therapy.”

  1. Line 186 - 189 this is a conclusion rather than a presentation of the result, so I would suggest to move this in the discussion. 

Response: Thank you. As suggested, we have revised this sentence and moved to the discussion.

In this study, the results indicated that, in patients with cardiogenic pulmonary edema, HFNC therapy could better improve the RR, lactate levels, SPO2, and ABG parameters over time more than conventional oxygen therapy.

“In this study, we observed that several objective parameters including RR, lactate levels, SpO2, and ABG parameters (PaCO2, pH, PaO2, pH, HCO3 –, and SpO2) in the HFNC-treated group clinically improved over time compared with conventional oxygen therapy in patients with cardiogenic pulmonary edema.”

  1. Can the Authors provide details re treatment? Specifically, I think it would be important for the Author to clarify the average flow-rate/Fio2 with HFNT and the flow-rate and device for O2 during the study period. Currently, only the initial setting are provided. This is of the utmost importance to understand the impact of the results on blood gas analysis as well.  

Response: Thank you for your helpful comments. We agree with your comment.

In our treatment protocol, oxygen therapy was commenced by using a conventional nasal cannula at a flow rate >2 L/min in the conventional oxygen therapy group. The flow rate was continuously adjusted for the conventional nasal cannula or face mask to maintain an SpO2 of >93%. In the HFNC group, oxygen therapy was applied through large-bore binasal prongs and a heated humidifier (MR850, Fisher & Paykel Healthcare Limited, Auckland, New Zealand) with a flow rate of 45 L/min and a fraction of inspired oxygen (FiO2) of 1.0 at initiation (Optiflow, Fisher and Paykel Healthcare, Auckland, New Zealand). The FiO2 (range 21–100%) and flow rate (≤60 L/min) in the system were adjusted to maintain an SpO2 of >93%. Thus, we designed this study by adjusting the FiO2 and flow rate to obtain the goal of maintaining an SpO2 >93%. Therefore, we recorded the values of the FiO2 and flow rate that achieved the stated goal of an SpO2 >93%. Furthermore, an ABGA was performed on time as stipulated in the predetermined protocol. However, we could not compare the clinical implications between parameters on the ABGA and the FiO2 or flow rate. Thus, we have added these values and limitations to the results and discussion sections as shown below.

Results

“The flow rate that resulted in the achievement of the goal of SpO2 >93% was 4.36 ± 3.35 L/min in the conventional O2 therapy group whereas the flow rate and FiO were 47.58 ± 5.02 L/min and 57.39 ± 14.38 in the HFNC group.”

Limitations

“As the present study was conducted by adjusting the FiO2 and flow rate with the goal of maintaining an SpO2 >93%, we recorded FiO2 and flow rate that achieved the stated goal. An ABGA was conducted in accordance with the predetermined protocol. Therefore, we could not directly compare the clinical implications between ABGA parameters and FiO2 or flow rate in each device. Further studies are needed to validate the impact of ABGA parameters by adjusting the FiO2 or flow rate in each device for O2.”

  1. Discussion
    1. Line 269-274 - The Author comment on the need for further studies to assess the effect on longer term outcome, ICU admission and ETI rate. However, this is as a consequence of the lack of difference in this outcomes in their study, which, however, was not powered or designed to assess such outcomes. 

Response: Thank you. We have accordingly deleted this point.

Third, we identified significant beneficial changes in the HFNC-treated group through the evaluation of objective parameters that were measured at 30 and 60 minutes after randomization. However, the specified post-intervention observation period may be insufficient to ascertain the long-term efficacy of HFNC.”

  1. Line 283 - Can the Author clarify? In the result section it was stated the the echo was within 6 months. Also, an echo 60 minutes after intervention is discussed, but in the result section only the baseline echo on admission is presented. Can the Author clarify and comment? If the echo in the ED was performed after the intervention, this should not be presented as a baseline characteristic in Table 1 as it would be confusing. Furthermore, can the Author comment on how the HFNT could influence echocardiography results. Finally, the echo (both retrieval of the most recent one, and performing in ED) is not discussed in the methods section.  

Response: Thank you for your kind feedback. We apologize for the inadvertent confusion that has been caused. We have clarified this point in the intervention and results section of the manuscript as follows.

“In this study protocol, we could determine the baseline ejection fraction (EF) from within 6 months before the ED admission on the basis of the latest echocardiographic examination. We obtained the baseline EF observed within 1 month preceding the ED from all subjects. In addition, we obtained the EF on emergency echocardiography that was undertaken within 60 minutes after the intervention in the ED.”

“There were no major between-group differences in the cotreatments, baseline ejection fraction (EF) within 1 month before ED admission by the latest echocardiographic examination, and EF on emergency echocardiography conducted within 60 minutes after the intervention in the ED.”

Furthermore, as suggested, we have moved the EF values identified on the emergency echocardiography that was conducted within 60 minutes after the intervention from Table 1 to Table 2.

Unfortunately, we could not conduct an emergency echocardiography before the intervention and after the ED admission in the emergency setting. Therefore, we could not directly compare the echocardiographic results from before to after the intervention. However, we cannot ascertain how the HFNC therapy affects cardiac function on the basis of echocardiographic findings.

We obtained echocardiographic studies within 1 month of the ED admission as well as 60 minutes post intervention. However, this study could not completely clarify the positive or negative effects of positive end-expiratory pressure (PEEP) of HFNC in the HFNC-treated patients because we were unable to obtain the pre-intervention echocardiographic findings after ED admission. In order to clarify the benefits of HFNC, further prospective multicenter trials are required to validate the usefulness of HFNC in patients with cardiogenic pulmonary edema.

Furthermore, we reviewed several articles in the literature on how HFNC can affect the echocardiographic findings. No direct studies have been conducted to determine how HFNC can affect echocardiographic findings in patients with cardiogenic pulmonary edema. However, we were able to indirectly ascertain the effect of HFNC on echocardiographic parameters in these patients through other investigations.

On comparing the HFNC and face-mask oxygen deliveries, at a flow rate of 35 L/min, we found that the HFNC nasopharyngeal pressure increased to 2.7 ± 1.04 and 1.2 ± 0.76 cmH2O with the mouth closed and open, respectively, whereas it was almost zero with the face mask. With due consideration of the sex and body mass index (BMI), whether the mouth was closed or opened, and the flow rate, other authors have reported positive pharyngeal pressure with HFNC [27]. With the mouth closed, the pharyngeal pressure linearly increases with the flow rate. With the mouth open, even at a flow rate of 60 L/min, the pharyngeal pressure remained below 3 cmH2O [27]. In postoperative patients, as the inspiratory flow increased, the airway pressure increased: 1.52 ± 0.7, 2.21 ± 0.8, and 3.1 ± 1.2 cmH2O at flow rates of 40, 50, and 60 L/min, respectively [27]

The HFNC is, therefore, not entirely similar to the application of continuous positive airway pressure (CPAP), which aims to maintain a steady positive pressure during the entire breathing cycle [28]. The target of HFNC is to improve flow instead of pressure; therefore, the objective parameter to investigate the effect of HFNC should not be the measurement of pharyngeal pressure, but instead, the changes in the hemodynamic status and increase in lung aeration [28].

Roca et al. demonstrated that the application of HFNC significantly decreased median IVC inspiratory collapse to approximately 20% of from baseline in patients with a New York Heart Association Class III heart failure [22]. These changes in the IVC inspiratory collapse were reversible after HFNC withdrawal [22]. The respiratory rate was significantly reduced from 23 breaths per minute at baseline to 17 breaths per minute on HFNC with 20 lpm and 13 breaths per minute on HFNC with 40 lpm [22]. In contrast, no significant changes in other echocardiographic or clinical variables were documented [22].  

We have added this point in the limitations section of the manuscript.

“Fourth, as the rate of HFNC increases, so does the positive pharyngeal pressure. The application of HFNC was maintained below 3 cmH2O of positive pharyngeal pressure even at a flow rate of 60 L/min with the mouth open [27,33]. Roca et al. demonstrated that the application of HFNC significantly decreased the median IVC inspiratory pressure approximately 20% from the baseline in patients with New York Heart Association Class III heart failure [22]. These changes in the IVC inspiratory collapse were reversible after HFNC withdrawal [22]. Nevertheless, there were no significant changes in other echocardiographic or clinical variables [22]. Although we obtained the data from echocardiographic studies conducted within 1 month before the ED admission and 60 minutes after the intervention, to compare the pre- and post-intervention parameters, this study could not completely clarify the positive or negative effects of positive end-expiratory pressure (PEEP) of HFNC in HFNC-treated patients because we were unable to obtain the echocardiographic results immediately before the intervention and after the ED admission.”

  1. The Authors mention the role of HFNT as a bridge prior to other noninvasive forms of support or to invasive ventilation. Can the Author discuss this more in depth? 
  2. I would like to read a more in depth discussion on how the proposed physiological mechanisms of HFNT lead to a beneficial effect and support the Author's results. 

Response: Thank you. We have accordingly revised the manuscript.

“Many previous studies of HFNC use in patients with acute respiratory failure have been conducted, although few studies have demonstrated the clinical effectiveness of HFNC in HF patients with cardiogenic pulmonary edema. The application of HFNC significantly improved oxygenation and decreased the RR in patients with respiratory failure [20]. A retrospective study by Jeong et al. demonstrated that the use of HFNC could significantly increase the PaO2, pH, and SpO2 and decrease the PaCO2 and RR in patients with hypercapnia in the ED [24]. In HFNC, the high flow washes out carbon dioxide from the anatomical dead space of nasopharynx and overcomes the resistance against expiratory flow [19,20,27,28]. This produces positive pressure within the nasopharyngeal space that is appropriate for recruiting the collapsed alveoli or for increasing the lung volume (CPAP effect) despite of its relatively low compared with closed systems [20,27,28]. HFNC maintains a relatively constant FiO2 because of the small difference between the high-flow oxygen that is delivered and the patient’s inspiratory flow [20,27,28]. Patients feel comfortable, and the mucociliary function remains good because HFNC can warm and humidify high flow [19,20,27,28]. Given the physiological benefits of HFNC, it is clear that HFNC provides a constant FiO2 and O2 with the nasal cannula, thereby reducing CO2 rebreathing, ensuring constant positive pressure, and providing a fresh O2 reservoir by washing out the nasopharyngeal dead space. Consequently, HFNC can maintain sufficient oxygenation by improving the respiratory load and gas exchange in cardiogenic pulmonary edema [23,29]. We found that HFNC could deliver effective oxygenation without major complications or life-threatening adverse events.”

Reviewer 2 Report

INTRODUCTION LINE 66: HFNC is not a ventilation method, it exerts a minimal wash-out action on the dead airway space, leading to a slight reduction in CO2 (please cite Eur J Intern Med. 2019 Jun;64:10-14. doi: 10.1016/j.ejim.2019.04.010)

Exclusion criteria: non-cardiogenic pulmonary edema (ARDS) and viral pneumonia were not included.

"O2 supply alone not being sufficient, and the need of immediate invasive trachea management due to severity of symptoms"; what were the criteria that led to immediate intubation? What type of respiratory failure has it been decided to treat? (pure hypoxemic or hypoxemic / hypercapnic?) in the results it is highlighted that the patients tended to be normocapnic.

A distinction between patients with chronic and non-chronic lung diseases (COPD, fibrosis) was not considered.

The study outcomes section should be included after the introduction;

"All-cause mortality within 28 days of ED admission": how was this parameter assessed? (medical records? check-ups? calls?)

Line 169: “Reasons for non-enrollment included end-stage renal disease, respiratory failure, concurrent pneumonia, myocardial infarction, and depressed consciousness”: pulmonary edema is a cause of respiratory failure so it cannot be used in general as an exclusion criterion ; pneumonia is included in this part of the text as an exclusion criterion, which was not stated in the exclusion criteria.

Baseline patient data: there is a significant difference between the pro-bnp values ​​in the two groups. Can the authors discuss this data?

It would have been fundamental to report parameters such as P / F and alveolar-arterial delta.

Results: COT and HFNC displayed the same intubation rate and multiple ICU accesses for HFNC, although not statistically significant.

Line 150: HFNC is not a ventilation, please re-write the sentence.

The authors conclude: "The application of HFNC could be considered as a choice of treatment for initial oxygen therapy in patients with cardiogenic pulmonary edema in the ED". These conclusions cannot be drawn given the lack of some key points in the planning of the work; first, the study should include a comparison with the CPAP that is currently gold standard in the treatment of cardiogenic pulmonary edema. Secondly, there is a lack of baseline distinction criteria for the patient (hypercapnic vs normocapnic) and clear early intubation criteria (HFNC cannot be the method of choice in the hypercapnic patient with altered pH). Therefore the work is limited to illustrating how by comparing COT vs HFNC at 1 hour there is a reduction of some blood gas and FR parameters, not leading to significant clinical variations (intubation rates, admission to ICU).

Author Response

**Please, check attached file **

Response to reviewers’ comments

Benefits of high flow nasal cannula therapy for acute pulmonary edema in patients with heart failure in the emergency department: a prospective multi-center randomized controlled trial

Comments and Suggestions for Authors

Response: Thank you for the time and effort you have expended to review our manuscript. We appreciate your kind comments and useful suggestions with regard to our manuscript.

1.INTRODUCTION LINE 66: HFNC is not a ventilation method, it exerts a minimal wash-out action on the dead airway space, leading to a slight reduction in CO2 (please cite Eur J Intern Med. 2019 Jun;64:10-14. doi: 10.1016/j.ejim.2019.04.010)

Response: Thank you for your kind comments. We have deleted this point on ventilation from the revised manuscript. In deference to your suggestion, we have cited the following reference in the revised manuscript (Andrea Boccatonda, Paolo Groff. High-flow Nasal Cannula Oxygenation Utilization in Respiratory Failure. Eur J Intern Med. 2019 Jun;64:10-14.)

The use of HFNC may be limited in the patient affected by hypercapnic respiratory failure because HFNC has a minimal effect on reducing the CO2 levels by a washout of the anatomical dead space [19]. In recent years, high flow nasal cannula (HFNC) therapy has been used as an effective approach of delivering sufficient oxygen to patients with acute respiratory failure because this device can potentially generate positive airway pressure, decrease entrainment of ambient air, and reduce the work of breathing [2,19]. Despite the patient’s discomfort with high-flow oxygen applications, the HFNC system can enhance the comfort and tolerability in patients, by integrating additional functions for humidification and warming of high-flow oxygen [2,19-21]. Based on the aforementioned characteristics, the use of an appropriate oxygen therapy can reduce the rate of intubation and mechanical ventilation in the ED  [8]

2.Exclusion criteria: non-cardiogenic pulmonary edema (ARDS) and viral pneumonia were not included.

Response: Thank you for your kind comment. We apologize for the inadvertent confusion that has been caused. During the screening process, we excluded patients with non-cardiogenic pulmonary edema (ARDS) and pneumonia based on the clinical manifestations, diagnostic criteria, and modalities. As suggested, we have included this point in the revised manuscript.

3."O2 supply alone not being sufficient, and the need of immediate invasive trachea management due to severity of symptoms"; what were the criteria that led to immediate intubation?

Response: To clarify this point, we have added the predetermined criteria for the early termination within the revised manuscript.

“However, according to predetermined criteria of early termination, early intubation and escalation of other devices was allowed if the patients had an intolerable response to the sustained oxygen therapy with either the conventional nasal cannula or HFNC. The early termination criteria included the failure to tolerate the therapy (respiratory rate >35 breaths/min, SpO2 <90%, PaO2/FiO2 <200 mmHg, pulse rate >120 beats/min or a >30% increase above the baseline, and a noninvasively measured pre-intervention mean arterial pressure that was >30% higher than that at the baseline, or signs of respiratory distress (e.g. tachypnea, use of accessory muscles of respiration, and abdominal paradox), and clinician judgements (when immediate intervention is required due to worsening of the levels of anxiety, agitation, and consciousness compared to those at the pre-intervention timepoint). If one or more of the early termination criteria were met, the oxygen therapy was escalated toward noninvasive ventilation or converted directly to intubation for mechanical ventilation.”

Results

There was no significant difference in the rates of endotracheal intubation within 24 hours in patients receiving conventional oxygen therapy (n=1, 3.0%) and those undergoing HFNC (n=1, 2.94%, p=0.999). All endotracheal intubations were undertaken after the completion of the study interventions. Therefore, patients were not excluded from this study merely due to the need for intubation.

What type of respiratory failure has it been decided to treat? (pure hypoxemic or hypoxemic / hypercapnic?) in the results it is highlighted that the patients tended to be normocapnic.

A distinction between patients with chronic and non-chronic lung diseases (COPD, fibrosis) was not considered.

Response: Thank you for your kind feedback. The study participants were enrolled on the basis of the SpO2 level; therefore, it was appropriate for the study to be conducted in patients with pure hypoxemic respiratory failure. Patients with fibrosis of the lung were excluded on the basis of findings on radiography, and the patients was not identified from the radiologist’s official report of the x-ray findings. In this study, we included the mental status as an exclusion criteria to facilitate the exclusion of patients with CO2 narcosis. The application of HFNC in patients with COPD had a positive effect on CO2 washout although it was minimal. HFNC has the advantage of constantly controlling FiO2. As the FiO2 with HFNC was not maintained at a higher level in patients with SpO2 >93% despite the abovementioned benefit, we believe that the application if HFNC has less efficacy than conventional O2 therapy in patients with CO2 retention.

4.The study outcomes section should be included after the introduction;

"All-cause mortality within 28 days of ED admission": how was this parameter assessed? (medical records? check-ups? calls?)

Response: Thank you for your kind suggestion. We reviewed the medical data from patient discharge and follow-up in the outpatient department that were recorded in the electronic health records (EHR) during the study period.

5.Line 169: “Reasons for non-enrollment included end-stage renal disease, respiratory failure, concurrent pneumonia, myocardial infarction, and depressed consciousness”: pulmonary edema is a cause of respiratory failure so it cannot be used in general as an exclusion criterion ; pneumonia is included in this part of the text as an exclusion criterion, which was not stated in the exclusion criteria.

Response: Thank you for your this valuable feedback. We have clarified this point in the manuscript and in the figure.

In this study, the patients with pneumonia were excluded during the screening process on the based on the possibility of pneumonia as evidenced in the past medical history, clinical manifestations, and unilateral consolidation on radiographic images. In accordance with the predetermined screening criteria, we confirmed patients with cardiogenic pulmonary edema as those with eligibility for study inclusion. However, patients with hypoxia due to pneumonia were excluded from this study.

6.Baseline patient data: there is a significant difference between the pro-bnp values ​​in the two groups. Can the authors discuss this data?

Response: Thank you for your kind suggestion. Accordingly, we have added this point to the discussion section in the manuscript.

Nonetheless, it is noteworthy that the Brain Natriuretic Peptide (BNP) value was statistically higher in the HFNC group than in the conventional O2 group in this study. BNP is a hormone that is secreted by the ventricle in response to increased ventricular volume or pressure [31]. The BNP value is elevated when left ventricular systolic function decreases, and this elevation is proportional to the severity of HF, according to the NYHA classification, and can indicate the long-term prognosis in patients with heart failure [32]. As the pro-BNP value was significantly higher in the HFNC group than in the conventional O2 group, the HFNC group may have a greater degree of severity of heart failure at the time of ED admission. However, in this study, we cannot exclude the possibility that the pro-BNP level could have affected clinical outcomes such as ICU admission and 28-day mortality. Moreover, it was difficult to conduct a cardiopulmonary exercise test and immediate echocardiography upon ED admission in the emergency setting of our study and, therefore, further prospective multicenter studies are required to validate the clinical utility of HFNC based on the HF severity of patients with cardiogenic pulmonary edema.

7.It would have been fundamental to report parameters such as P / F and alveolar-arterial delta.

Results: COT and HFNC displayed the same intubation rate and multiple ICU accesses for HFNC, although not statistically significant.

Response: Thank you for your insightful feedback. Accordingly, we have now added results of the P/F to Table. Unfortunately, we did not collect the data on the alveolar–arterial delta during the prospective study and, as we cannot newly review these data because of the use of anonymized patient data in this study, we are unable to report the alveolar-arterial delta as a parameter in this study. We apologize for this drawback.

8.Line 150: HFNC is not a ventilation, please re-write the sentence.

Response: Thank you for your kind feedback. We have accordingly revised this point in the manuscript.

“We found that HFNC could deliver effective oxygenation and adequate ventilation without major complications or life-threatening adverse events.”

Conclusions

Compared with conventional oxygen therapy, HFNC could significantly improve several objective parameters over time such as RR, lactate levels, and ABG reflection of oxygenation and ventilation, after ED admission in HF patients with acute pulmonary edema.

The authors conclude: "The application of HFNC could be considered as a choice of treatment for initial oxygen therapy in patients with cardiogenic pulmonary edema in the ED". These conclusions cannot be drawn given the lack of some key points in the planning of the work; first, the study should include a comparison with the CPAP that is currently gold standard in the treatment of cardiogenic pulmonary edema. Secondly, there is a lack of baseline distinction criteria for the patient (hypercapnic vs normocapnic) and clear early intubation criteria (HFNC cannot be the method of choice in the hypercapnic patient with altered pH). Therefore the work is limited to illustrating how by comparing COT vs HFNC at 1 hour there is a reduction of some blood gas and FR parameters, not leading to significant clinical variations (intubation rates, admission to ICU).

Response: Thank you for your kind comment. We completely concur with your opinion. Accordingly, we have revised this point in conclusion of the manuscript as shown below.

“The application of HFNC could replace conventional O2 therapy as an initial effective oxygen therapy in patients with cardiogenic pulmonary edema in the ED."

Round 2

Reviewer 2 Report

AUTHORS HAVE ADEQUATELY RESPONDED TO THE OBSERVATIONS OF THE REVIEWERS